# Evaluation of Binding and Neutralizing Antibodies for Inactivated SARS-CoV-2 Vaccine Immunization

**DOI:** 10.3390/diseases12040067

**Published:** 2024-03-28

**Authors:** Heng Zhao, Guorun Jiang, Cong Li, Yanchun Che, Runxiang Long, Jing Pu, Ying Zhang, Dandan Li, Yun Liao, Li Yu, Yong Zhao, Mei Yuan, Yadong Li, Shengtao Fan, Longding Liu, Qihan Li

**Affiliations:** Key Laboratory of Systemic Innovative Research on Virus Vaccine, Institute of Medical Biology, Chinese Academy of Medical Sciences and Peking Union Medical College, Kunming 650118, China; zhaoheng@imbcams.com.cn (H.Z.); jgr@imbcams.com.cn (G.J.); lc2011@imbcams.com.cn (C.L.); cheyanchun@imbcams.com.cn (Y.C.); runxianglong@imbcams.com.cn (R.L.); pujing@imbcams.com.cn (J.P.); zhangy@imbcams.com.cn (Y.Z.); lidandan@imbcams.com.cn (D.L.); liaoyun@imbcams.com.cn (Y.L.); yuli@imbcams.com.cn (L.Y.); zhaoyong@imbcams.com.cn (Y.Z.); yuanmei@imbcams.com.cn (M.Y.); yadong_li@imbcams.com.cn (Y.L.); liuld@imbcams.com.cn (L.L.)

**Keywords:** COVID-19 diagnosis, binding antibody, neutralization antibody, evaluation

## Abstract

The circulating severe acute respiratory syndrome coronavirus-2 (SARS-CoV-2) variant presents an ongoing challenge for surveillance and detection. It is important to establish an assay for SARS-CoV-2 antibodies in vaccinated individuals. Numerous studies have demonstrated that binding antibodies (such as S-IgG and N-IgG) and neutralizing antibodies (Nabs) can be detected in vaccinated individuals. However, it is still unclear how to evaluate the consistency and correlation between binding antibodies and Nabs induced by inactivated SARS-CoV-2 vaccines. In this study, serum samples from humans, rhesus macaques, and hamsters immunized with inactivated SARS-CoV-2 vaccines were analyzed for S-IgG, N-IgG, and Nabs. The results showed that the titer and seroconversion rate of S-IgG were significantly higher than those of N-IgG. The correlation between S-IgG and Nabs was higher compared to that of N-IgG. Based on this analysis, we further investigated the titer thresholds of S-IgG and N-IgG in predicting the seroconversion of Nabs. According to the threshold, we can quickly determine the positive and negative effects of the SARS-CoV-2 variant neutralizing antibody in individuals. These findings suggest that the S-IgG antibody is a better supplement to and confirmation of SARS-CoV-2 vaccine immunization.

## 1. Introduction

In the realm of research focusing on vaccines against COVID-19 caused by the novel coronavirus, a diversity of vaccine types has been explored. These include inactivated virus vaccines, recombinant protein vaccines, mRNA and vector vaccines, etc. [1,2]. The design of these vaccines capitalizes on the structural characteristics of the virus and exposes the spike (S) protein located on the virus’s surface as the primary antigenic target [3,4,5], aiming to induce a specific immune response capable of neutralizing the invading viruses [6,7]. At the same time, immunological analysis targeting the N protein, which is a major structural component of SARS-CoV-2, suggests that this antigen can also induce specific antibody and cytotoxic T lymphocyte (CTL) responses in infected individuals [8]. Some vaccines, including mRNA and DNA vaccines, use the N protein as the main antigenic target [9]. 

For evaluating the efficacy of COVID-19 vaccines, rhesus macaques and Syrian hamsters are the most common animal models used to evaluate vaccines [10], while neutralizing antibodies (Nabs) are important indicators of vaccines. However, virus neutralization can only be tested in high-level biosafety laboratories, requiring much time and effort. Thus, describing the relativity between binding antibodies (such as S-IgG and N-IgG) and Nab titers is significant for evaluating vaccines and screening vaccine candidates. So far, many studies on the detection of S-IgG and N-IgG, as well as on the comparison between IgG and Nabs, have been reported [11,12], but in most of these studies, only the correlation between IgG and Nab has been evaluated, and the value of an IgG titer in determining whether a Nab is positive has not been evaluated further. In addition, there has been no quantification of the value, such as how high the IgG titer is (the threshold), to determine whether the corresponding Nab is positive and how accurate this prediction is, which is considered critical in vaccine and epidemiological investigations on COVID-19 because ELISA can detect IgG in a large number of serum samples in a rapid and high throughput compared with neutralization experiments. More importantly, a neutralization experiment must be conducted in a laboratory with a biosafety level of 3 or above, a huge obstacle for general laboratories. To better understand the relationship between S-IgG, N-IgG, and Nabs, to more effectively solve the problem of how to judge whether the Nab is positive or not according to an IgG titer, and to combine IgG titers with Nabs scientifically, we try to analyze the relationship between S-IgG, N-IgG, and Nabs by using the receiver operating characteristic curve (ROC curve). Although ROC curves are mostly used to evaluate the diagnostic effect of discriminant models [13], using ROC curves for analysis is scientific and feasible in IgG detection. 

In this study, we detected the S-IgG and N-IgG titers in sera collected from humans, rhesus macaques, and hamsters immunized with the COVID-19 inactivated vaccine and performed a comprehensive analysis of Nabs. Compared with the receiver operating characteristic curves (ROCs) of correlation between S-IgG (or N-IgG) and Nabs, we considered the measurement of the area under the ROC curve (AUC) and determined the potency threshold of S-IgG and N-IgG in judging the positive conversion of Nabs. These findings suggested that the S-IgG antibody is a better supplement to and confirmation of inactivated SARS-CoV-2 vaccine effectiveness.

## 2. Materials and Methods

### 2.1. Cell and Virus

The Vero cell used for the Nab test was obtained from the Institute of Medical Biology, Chinese Academy of Medical Sciences (IMBCAMS). The SARS-CoV-2 virus used for the Nab test (Wuhan strain KMS-1, MT226610.1) was isolated from the Yunnan Provincial Infectious Disease Hospital and stored by the IMBCAMS. The S1 antigen and N antigen (Wuhan strain) were purchased from San You Biomedical Co., Ltd. (Shanghai, China).

### 2.2. Inactivated Vaccine Production and Immunization

The strain KMS-1 was inoculated into Vero cells and inactivated with formaldehyde (*v*/*v* = 1:4000) for 48 h. After inactivation, the virus was purified and concentrated. Subsequently, a second inactivation was performed using beta-propiolactone (*v*/*v* = 1:2000) on the initially inactivated viral fluid, followed by purification and concentration to obtain the vaccine stock. ELISA was employed for antigen quantification of the vaccine stock. Based on the antigen content, aluminum hydroxide (Al(OH)_3_) was added to achieve a final concentration of 0.25 mg/mL, and the mixture was adsorbed to prepare the vaccine (100U/0.5 mL/dose).

The vaccine was inoculated into humans, rhesus macaques, and hamsters intramuscularly. Healthy volunteers aged 18–59 were recruited for the clinical trials, which were conducted according to the principles of randomization, double-blinding, and placebo control. Blood samples were taken from the volunteers as a baseline for the first time. In the vaccine group, each volunteer received an intramuscular injection of one dose (100U/0.5 mL/dose) of the inactivated SARS-CoV-2 vaccine as the first immunization. A booster immunization was given 14 days after the first immunization. Blood samples were taken from the volunteers on day 14 after the booster immunization to evaluate the immunization effects, including neutralizing antibodies, cell-mediated immunity, and IgG levels. The immunization program for rhesus macaques and hamsters was consistent with that of the clinical trial. Clinical sera (n = 297), rhesus macaque sera (n = 35), and hamster sera (n = 62) were collected for the detection of S-IgG, N-IgG, and Nabs. In the human cohort, all the healthy volunteers were excluded from natural infections before enrollment. Ethical approval for preclinical animal experiments was obtained from the Ethics Committee of the Institute of IMBCAMS (approval number: DWSP202003005). The clinical trials were approved by the West China Second University Hospital, Sichuan University Ethics Committee (approval number: Y2020008).

### 2.3. Neutralizing Antibody Detection

The neutralization experiment was completed in a biosafety level 3 (BSL-3) laboratory of IMBCAMS. The serum samples were two-fold diluted to 1:4–1:512, 50 µL was added to 96-well cell culture plates for each dilution, and each well was supplemented with 50 µL of COVID-19 (100 CCID50/well). The plates were placed in a 37 °C incubator for neutralization for 2 h. Then, 100 µL of Vero cell suspension (80,000–100,000 cells/mL) was added to each well. Each plate was placed in a 5% CO_2_ incubator at 37 °C for 7 days. The neutralization antibody titer was determined according to the cytopathic effect (CPE) observed in the wells at each dilution. The Nab titers were recorded as reciprocal of the highest serum dilutions without CPE observed in the well (titers < 1:4 were considered negative); if the test result of the sample is negative, the titer of Nab is defined as 1. Each species’ GMT (Geometric Mean Titer) of Nab was calculated for the positive samples.

### 2.4. ELISA Detection

ELISAs were performed with antibodies against the S1 and the N that were developed by this institute. Briefly, S1 protein and N protein were diluted with 0.01 mol/L PBS to final concentrations of 1 µg/mL and 2 µg/mL, respectively. The diluted S1 protein and N protein were added to 96-well plates, 100 µL was added to each well, and the plates were placed at 4 °C for adsorption overnight. Then, the plates were washed once with 0.01 mol/L PBS, and then 200 µL of 1% BSA (Bovine serum albumin) solution was added to each well, and the plates were placed at 4 °C overnight for sealing. The next day, the sealing fluid was discarded, and the plates were dried at room temperature. Then, the plates were sealed in an aluminum plastic bag and stored at −20 °C for use.

### 2.5. Serum Dilution and Test

The serum was diluted to 1:400 with diluent (PBS containing 10%NBS) in advance, and then the sample (400 times diluted) was diluted 2-fold in series with diluent to 1:51,200 for detection. The serum samples of each dilution (from 1:400 to 1:51,200) were added to the S-IgG detection plate and N-IgG detection plate, with 100 µL for each well. At the same time, negative serum controls (1:400 dilution of healthy human serum/rhesus macaque serum/hamster serum) were used. Following incubation at 37 °C for 1 h, the plate was washed five times with washing buffer (PBS containing 0.05% Tween 20). Then, 100 µL of goat anti-human IgG-HRP for detecting human S-IgG and N-IgG was added to each well. Similarly, goat anti-hamster IgG-HRP for detecting hamster S-IgG and N-IgG was added to each well, and goat anti-rhesus macaque IgG-HRP for the detection of rhesus macaque S-IgG and N-IgG was added to each well and incubated at 37 °C for 0.5 h. After being washed five times, 100 µL of color development solution (TMB) was added to each well, and the color was developed in the dark at room temperature for 10 min. A stop solution was added to each well to stop the reaction (50 µL/well), and a microplate reader was used to read the absorbance value (OD value) of each well. With 450 nm as the absorption wavelength and 630 nm as the reference wavelength, the absorbance value at 450 nm minus the absorbance value at 630 nm is the final absorbance value of the sample. The 2.1-fold average OD value of the negative control was used as the cutoff value. When the OD value of the sample is greater than or equal to the cutoff value, it is positive; otherwise, it is negative. The reciprocals of highest dilution (OD ≥ cutoff) are defined as the titer of S-IgG or N-IgG antibody for the samples; if the test result of the sample is negative, the titer of IgG is defined as 1. Each species’ GMTs of S-IgG and N-IgG were calculated for the positive samples.

### 2.6. Statistics Analysis

The titers of S-IgG, N-IgG, and Nabs were drawn with GraphPad Prism 7.0 for Windows, and the GMTs of positive samples were compared using one-way analysis of variance (ANOVA) in SPSS 24.0. The positive rates of S-IgG and the coincidence rate of S-IgG, N-IgG, and Nab were calculated, and the differences between them were compared using a chi-square test in SPSS 24.0. In addition, the correlation between S-IgG, N-IgG, and Nab was analyzed using Spearman’s correlation analysis in GraphPad Prism 7.0. In addition, Nab results were used as the standard (control); ROC curves of S-IgG, N-IgG, and (S+N)-IgG were drawn using SPSS 24.0, and the area under the curve (AUC) and threshold (cutoff) were calculated according to the Jorden index (sensitivity+specitivity-1). When the Jorden index was the biggest, the corresponding value was taken as a threshold value, and the difference between AUCs was compared in MedCalc. *p* < 0.05 was considered statistically significant.

## 3. Results

### 3.1. Comparing Positive Rates between S-IgG, N-IgG, and Nabs of Humans, Rhesus Macaques, and Hamsters

Based on the ELISA results, the positive rate (number of positive cases/total number of cases × 100%) of S-IgG or N-IgG for each species was calculated. Similarly, according to the results of the neutralization test, the positive rate of Nabs for each species was calculated. According to the ELISA and neutralization results, the concordance rates of S-IgG or N-IgG with Nabs for each species were calculated. The results showed that the positive rates of S-IgG, N-IgG, and Nabs were 91%, 72%, and 95% in humans, 86%, 54%, and 91% in rhesus macaques, and 87%, 69%, and 90% in hamsters, respectively. Among them, the positive rate of Nabs in humans was the highest (95%). The comparative analysis showed significant differences between S-IgG and N-IgG and between N-IgG and Nabs (*p* < 0.001). The coincidence rates between the results of ELISA (the positive and negative of S-IgG or N-IgG) and Nabs suggested that S-IgG was more consistent with Nabs in humans, rhesus macaques, and hamsters. Further statistical analyses showed that the Nab-S-IgG coincidence rate was significantly higher than the Nab-N-IgG coincidence rate (*p* < 0.001) (Table 1).

### 3.2. Comparison between S-IgG and N-IgG in Humans and Evaluation of Correlations between S-IgG, N-IgG, and Nabs

The GMT values of IgG and Nab in human positive samples were calculated, and the difference between the S-IgG and N-IgG levels was compared. Additionally, the correlation between S-IgG and N-IgG was analyzed, as was the correlation between IgG and Nab. The results suggested that the vaccine induced higher S-IgG and N-IgG levels in humans, but the GMT of S-IgG was higher than that of N-IgG (Figure 1a). The correlation analysis showed that the correlations (r) between S-IgG and N-IgG, S-IgG and Nabs, and N-IgG and Nabs were 0.259, 0.624, and 0.246, respectively. Among them, the correlation between S-IgG and Nabs was the strongest; this was followed by the correlation between S-IgG and N-IgG (r = 0.259), and the weakest correlation was between N-IgG and Nabs. The results are shown in Figure 1a–d.

### 3.3. Comparison between S-IgG and N-IgG of Rhesus Macaques and Evaluation of Correlations between S-IgG, N-IgG, and Nabs

According to calculating and comparing the GMT of S-IgG, N-IgG, and Nab in rhesus macaques serums and analyzing the correlation between S-IgG and N-IgG, as well as between IgG and Nab, it is demonstrated that the inactivated SARS-CoV-2 vaccine induced rhesus macaques to produce high levels of S-IgG, N-IgG, and Nabs. Among these, the average level (GMT) of S-IgG was significantly higher than that of N-IgG, with a large statistical difference (Figure 2a). Further correlation analysis between S-IgG and N-IgG, S-IgG and Nab, and N-IgG and Nab showed that the strongest correlation was between S-IgG and Nab, with a correlation coefficient (r) of 0.765. This was followed by the correlation between N-IgG and Nab (r = 0.577). The weakest correlation was between S-IgG and N-IgG, with a correlation coefficient (r) of 0.359. The results are shown in Figure 2a–d.

### 3.4. Comparison between S-IgG and N-IgG of Hamsters and Evaluation of Correlations between S-IgG, N-IgG, and Nabs

The IgG and Nab titers of hamster serums were detected, compared, and analyzed by the same method as those of humans and rhesus macaques. It showed that higher levels (GMTs) of S-IgG, N-IgG, and Nabs were also induced in hamsters immunized with the inactivated SARS-CoV-2 vaccine. Similarly, the average level of S-IgG was significantly higher than that of N-IgG, showing a substantial statistical difference (Figure 3a). This was consistent with the results in humans and rhesus macaques. The correlation analysis also revealed that the strongest correlation was between S-IgG and Nabs, with a correlation coefficient (r) of 0.858. This was followed by the correlation between N-IgG and Nabs, with a coefficient (r) of 0.409. The weakest correlation was between S-IgG and N-IgG, with a correlation coefficient (r) of 0.262, as shown in Figure 3a–d. 

### 3.5. ROC Analysis of S-IgG and N-IgG Relating to Nabs in Humans, Rhesus Macaques, and Hamsters

The overall ROC curves were generated based on sensitivity versus (1-specificity) of S-IgG or N-IgG from different species. The ROC curve analysis showed that the AUCs (area under the curve) of S-IgG in humans, rhesus macaques, and hamsters were 0.89, 0.97, and 0.98, respectively, and the AUCs of N-IgG were 0.65, 0.80, and 0.84, respectively. When S-IgG and N-IgG were combined, their respective areas under the curve were 0.88, 0.98, and 1.00, respectively, which were not different from the respective areas under the curve of S-IgG (*p* > 0.05) (Figure 4 and Table 2). However, the area under the S-IgG and N-IgG curves of the same species had large and statistically significant differences (*p* < 0.05). The Jorden indexes were calculated, the maximum Jorden indexes were found, and the corresponding values were determined as the threshold. With the maximum Jorden index, the thresholds (cutoffs) of S-IgG of the three species (humans, rhesus macaques, and hamsters) were 600, 800, and 800, and the corresponding specificities (sensitivities) were 0.93 (0.71), 1.00 (0.94), and 1.00 (0.96), respectively. The thresholds of N-IgG were 600, 200, and 1200, and the corresponding specificities (sensitivities) were 0.53 (0.71), 1.00 (0.59), and 1.00 (0.59), respectively (Table 3).

## 4. Discussion

The S and N antigens of novel coronavirus (SARS-CoV-2) have demonstrated potent immunogenicity, prompting the production of specific IgG antibodies detectable via ELISA in the serums of vaccinated individuals or those recovering from COVID-19. Therefore, these two antigens, especially the S antigen, were widely used as antigen targets in vaccine design, such as in the currently marketed inactivated whole virus vaccine, the mRNA vaccine, and the submit vaccine. Clinical research has also shown that these vaccines can induce the body to produce S-IgG or (S+N)-IgG, but Nab is one of the indicators used to evaluate the effectiveness of the COVID-19 vaccine. Therefore, to further understand the relationships between S-IgG, N-IgG, and Nabs and to design and evaluate COVID-19 vaccines effectively, in this study, we compared and analyzed the relationships between S-IgG, N-IgG, and Nabs in the serums of humans, rhesus macaques, and hamsters who received the COVID-19 inactivated vaccine.

The results showed that the overall level of S-IgG was higher than that of N-IgG in the same species, with large differences. Among different species, the overall levels of S-IgG were successively distributed in hamsters, rhesus macaques, and humans, while the overall levels of N-IgG and Nabs were distributed in hamsters, humans, and rhesus macaques. In general, the overall levels of the three antibodies in hamsters were higher than those in humans and rhesus macaques, which may be related to the higher sensitivity of hamsters to COVID-19 [14]. The analysis of the positive rates of each antibody in the three species showed that the positive rates of S-IgG and Nabs in the three species were higher than those of N-IgG. There was no significant difference between the positive rates of S-IgG and Nabs, but there was a significant difference between them and the positive rates of N-IgG. These results were consistent with those of Wang Bing et al. [15]. The further comparison of the coincidence rates between S-IgG, N-IgG, and Nabs showed that there was little difference in the coincidence rate between S-IgG and Nab among the three species. Among them, the coincidence rate between S-IgG and Nabs in hamsters was the highest (97%), while the coincidence rate between N-IgG and Nabs was lower than that of S-IgG, and the difference was large. On the whole, the positive rate of S-IgG and the coincidence rates of S-IgG and Nabs were higher than those of N-IgG, which also reflected that the S antigen of the COVID-19 inactivated vaccine is dominant in stimulating antibody production and is the main immunogen, consistent with the results of Iyer [16]. 

Although the S-IgG and N-IgG immune responses occurred in the different species immunized by the COVID-19 inactivated vaccine, analysis of the correlation between S-IgG, N-IgG, and Nabs showed that the correlations between S-IgG and Nabs in the three species were higher than the correlations between N-IgG and Nabs, which is consistent with the results of Verena Krahling et al. [17]. Among these, the correlation between S-IgG and Nabs in hamsters was the highest (r = 0.858), followed by that of rhesus macaques and humans, suggesting that S-IgG may play a major role in neutralizing COVID-19; however, the role of N-IgG cannot be ignored. Although there appears to be an internal relationship between S-IgG or N-IgG and Nabs, especially S-IgG, which was characterized by a close correlation between S-IgG and Nabs and a high coincidence rate between S-IgG and Nabs, these internal relationships did not seem very clear. It is not possible to determine whether Nabs are positive based on the titers of S-IgG or N-IgG, which is very important for the evaluation of vaccine effectiveness and epidemiological research, as Nabs are the key indicators for evaluating vaccine effectiveness as well as for the evaluation of the immune protection effectiveness of vaccine recipients or recoveries. However, Nab testing needs to be conducted in a laboratory with a biosafety level of 3; this is a huge challenge for the general laboratory. To further understand the relationships between S-IgG, N-IgG, and Nabs, as well as quantify the relationship between IgG and Nabs, and more effectively solve the problem of determining whether Nabs are positive based on IgG titers, we used ROC curves to further analyze the relationships between S-IgG, N-IgG, and Nabs. The results showed that in the three species (human, rhesus macaque, and hamster), the areas under the ROC curve of S-IgG (AUC) were higher than those of N-IgG; the AUCs of S-IgG were 0.89, 0.97, and 0.98, respectively, and those of N-IgG were 0.65, 0.80, and 0.84, respectively. In addition, the AUC showed no change compared to S-IgG after combining S-IgG and N-IgG (S-IgG + N-IgG). A statistical analysis showed that the AUC of S-IgG was significantly different from that of N-IgG, which may result from differences in the immunogenicity of the S and N antigens in inactivated vaccines. However, there were no significant differences in the AUCs of S-IgG among the three species. The area under the ROC curve of S-IgG in humans was consistent with the results of Livia Mazzin et al. [18] (AUC = 0.90). In addition, we also determined the threshold values and the corresponding specificity and sensitivity of S-IgG and N-IgG in determining the positive conversion of Nabs in three species (humans, rhesus macaques, and hamsters). When thresholds of S-IgG for the three species reach 600, 800, and 800, respectively, the accuracy of judging the positive conversion of Nabs can reach 89–98%; however, when the thresholds of N-IgG for the three species reach 600, 200, and 1200, respectively, the accuracy of judging the positive conversion of Nabs can reach 65–84% only. It can be concluded that S-IgG of the three species is superior to N-IgG in judging whether Nab is positive, but there was no difference in the role of S-IgG among different species in judging whether Nab is positive.

## 5. Conclusions

The ELISA method allows for rapid and high-throughput detection of S-IgG titers, enabling accurate determination of Nab in hamsters, followed by rhesus macaques and humans. This study demonstrated that the S-IgG antibody is a better supplement to and confirmation of inactivated SARS-CoV-2 vaccine immunization evaluation. 

## Figures and Tables

**Figure 1 diseases-12-00067-f001:**
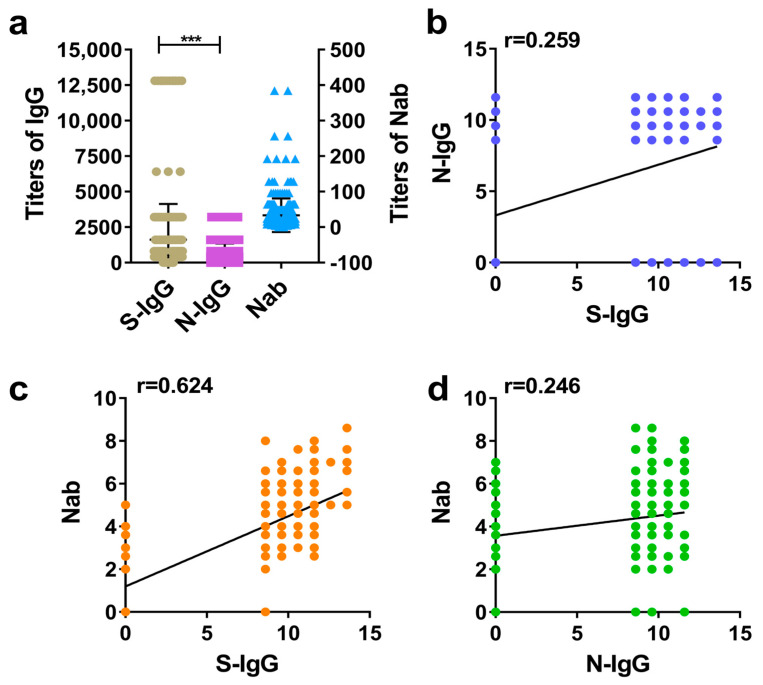
Comparative analysis of S-IgG, N-IgG, and Nabs in the serums of humans. (**a**) Distribution and comparison of S-IgG, N-IgG, and Nab; brown dots represent S-IgG titers, purple squares represent N-IgG titers, and blue triangles represent Nab titers. (**b**) Correlation between S-IgG and N-IgG; the dark blue dots represent the correlation between S-IgG and N-IgG. (**c**) Correlation between S-IgG and Nabs; the orange dots represent the correlation between S-IgG and Nabs. (**d**) Correlation between N-IgG and Nabs; the green dots represent the correlation between N-IgG and Nabs. Correlation analysis was performed after taking the logarithm (log2) of S-IgG, N-IgG, and Nab. “***” *p*-value < 0.001.

**Figure 2 diseases-12-00067-f002:**
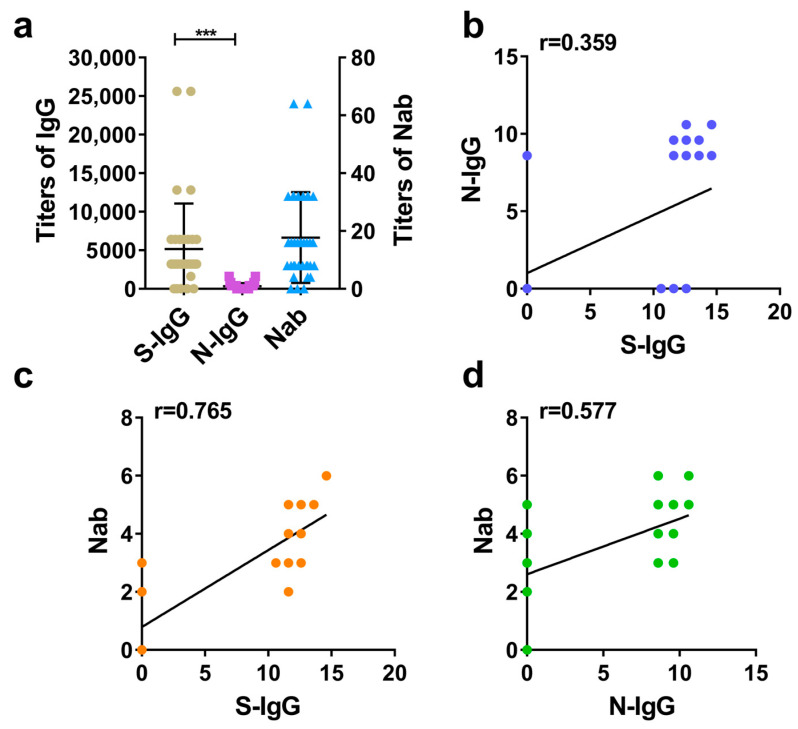
Comparative analysis of S-IgG, N-IgG, and Nabs in the serums of rhesus macaques. (**a**) Distribution and comparison of S-IgG, N-IgG, and Nab; brown dots represent S-IgG titers, purple squares represent N-IgG titers, and blue triangles represent Nab titers. (**b**) Correlation between S-IgG and N-IgG; the dark blue dots represent the correlation between S-IgG and N-IgG. (**c**) Correlation between S-IgG and Nabs; the orange dots represent the correlation between S-IgG and Nabs. (**d**) Correlation between N-IgG and Nabs; the green dots represent the correlation between N-IgG and Nabs. Correlation analysis was performed after taking the logarithm (log2) of S-IgG, N-IgG, and Nab. “***” *p*-value < 0.001.

**Figure 3 diseases-12-00067-f003:**
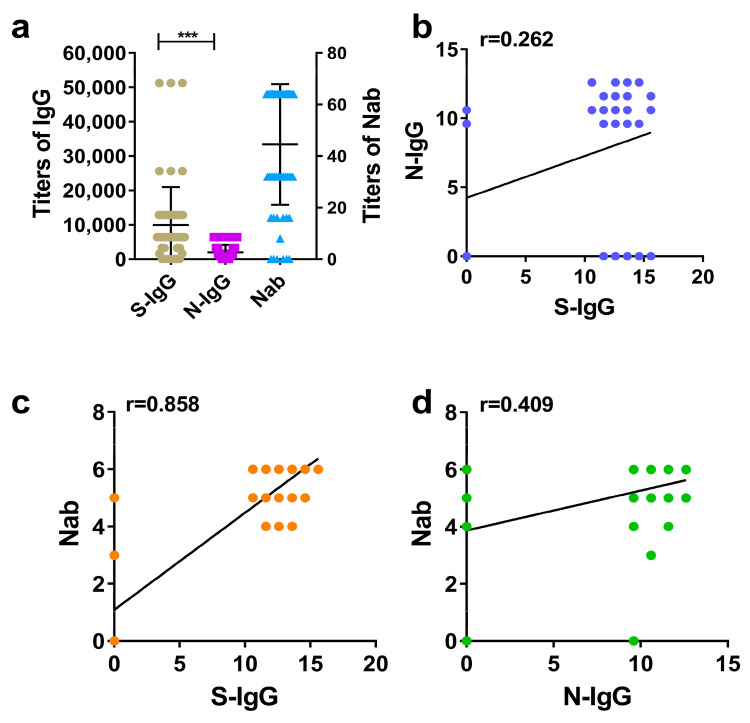
Comparative analysis of S-IgG, N-IgG, and Nabs in the serums of hamsters. (**a**) Distribution and comparison of S-IgG, N-IgG, and Nab; brown dots represent S-IgG titers, purple squares represent N-IgG titers, and blue triangles represent Nab titers. (**b**) Correlation between S-IgG and N-IgG; the dark blue dots represent the correlation between S-IgG and N-IgG. (**c**) Correlation between S-IgG and Nabs; the orange dots represent the correlation between S-IgG and Nabs. (**d**) Correlation between N-IgG and Nabs; the green dots represent the correlation between N-IgG and Nabs. Correlation analysis was performed after taking the logarithm (log2) of S-IgG, N-IgG, and Nab. “***” *p*-value < 0.001.

**Figure 4 diseases-12-00067-f004:**
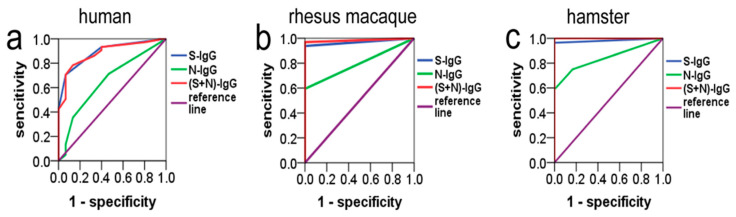
ROC curves of S-IgG, N-IgG, and (S+N)-IgG of three species. The neutral antibodies (positive and negative) detected in the neutralization experiment were used as a standard control, and the true positive rates (TPRs: sensitivity) and false positive rates (FPRs: 1-specificity) of the S-IgG and N-IgG tested using ELISA were calculated according to the control. ROC curves were drawn with TPR as the Y-axis and FPR as the X-axis using the SPSS 24.0 software. The blue curves represent S-IgG, the green curves represent N-IgG, and the red curves represent (S+N)-IgG. (**a**) S-IgG, N-IgG, and the combination of two IgG (S-IgG+N-IgG) ROC curves of humans. (**b**) S-IgG, N-IgG, and the combination of two IgG (S-IgG+N-IgG) ROC curves of rhesus macaques. (**c**) S-IgG, N-IgG, and the combination of two IgG (S-IgG+N-IgG) ROC curves of hamsters.

**Table 1 diseases-12-00067-t001:** Positive rates and consistency rates of S-IgG, N-IgG, and Nab in three species.

Species	Positive Rate (%)			Consistency Rate (%)	
	S-IgG	N-IgG	Nab	(S-IgG)-Nab	(N-IgG)-Nab
human	91 (269/297)	72 (213/297)	95 (282/297)	92 (274/297)	70 (209/297)
rhesus macaque	86 (30/35)	54 (19/35)	91 (32/35)	94 (33/35)	63 (22/35)
hamster	87 (54/62)	69 (43/62)	90 (56/62)	97 (60/62)	76 (47/62)

**Table 2 diseases-12-00067-t002:** Areas under the ROC curve of three species.

Antibody Type	Human					Rhesus Macaque					Hamster				
	AUC	SE	*p*	95% CI		AUC	SE	*p*	95% CI		AUC	SE	*p*	95% CI	
				Lower	Upper				Lower	Upper				Lower	Upper
S-IgG	0.89	0.03	0.00	0.83	0.96	0.97	0.03	0.08	0.91	1.00	0.98	0.02	0.00	0.95	1.00
N-IgG	0.65	0.07	0.05	0.51	0.79	0.80	0.09	0.09	0.61	0.97	0.84	0.06	0.01	0.73	0.95
S + N-IgG	0.88	0.04	0.00	0.81	0.96	0.98	0.20	0.01	0.94	1.00	1.00	0.00	0.00	1.00	1.00

Note: AUC: area under the curves; SE: standard error; 95% CI: 95% confidence interval.

**Table 3 diseases-12-00067-t003:** Area, threshold, specificity, and sensitivity of S-IgG and N-IgG curves of three species.

Species	S-IgG				N-IgG			
	AUC	Cutoff	Specificity	Sensitivity	AUC	Cutoff	Specificity	Sensitivity
human	0.89	600	0.93	0.71	0.65	600	0.53	0.71
rhesus macaque	0.97	800	1.00	0.94	0.80	200	1.00	0.59
hamster	0.98	800	1.00	0.96	0.84	1200	1.00	0.59

## Data Availability

Data will be made available on request.

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
