# Peer review of "Evaluation of Binding and Neutralizing Antibodies for Inactivated SARS-CoV-2 Vaccine Immunization"

_diseases, 2024, doi:10.3390/diseases12040067_

Round 1
Reviewer 1 Report
Comments and Suggestions for Authors
This article has a very interesting rationale, but unfortunately it is written in poorly understandable English and often, even if you understand what the authors mean, it is not clear.
For example in line 37-38, it is not clear to me what the authors mean and the reference does not help me, as 2 works are indicated (both in point 9) but neither of them is relevant
In the materials and methods section, paragraph 2.1 should be rewritten, better explaining in which type of assay and why the cells, viurs and other reagents are used
Since these are studies on human sera obtained from an unregistered vaccine, the authors should also mention the clinical trial and the procedures followed in detail in 2.2 section.
In the lane 96, what does p.i.stand for? Via the peritoneal inoculum? Also for humans?
The results section is confusing, before saying the result the experiment should be briefly introduced.
The order: humans, rhesus macaques and hamsters, in the text, figures and tables should always be the same, to maintain the same logical thread.
In line 199 is r=0.577 right?
In line 235 is 0.96 right?
Rewrite the conclusions better, making sure to specify which species we are talking about
Reviewer 2 Report
Comments and Suggestions for Authors
This study serves as a foundational investigation to determine the relationship between S-IgG, N-IgG, and neutralizing antibodies (Nabs) following inactivated SARS-CoV-2 vaccination in three species: humans, rhesus macaques, and hamsters. The results convincingly demonstrate that the correlations between S-IgG and Nabs were stronger than those between N-IgG and Nabs. It is noteworthy that N-IgG levels correspond to those seen in natural infections. In this context, it would be beneficial to include a brief description of whether the human cohort in this study had experienced natural infection prior to sample collection. Additionally, comparing this cohort with one that received mRNA vaccinations during the same period would provide valuable insights, although its absence represents a limitation of this study. Furthermore, the conclusion section and the final sentence of the abstract should clearly indicate that the results were obtained from immunization with an inactivated vaccine. This clarification is important to avoid confusion with the effects of mRNA vaccination. Future studies exploring the variance in antibody responses across different patient characteristics are warranted.
Reviewer 3 Report
Comments and Suggestions for Authors
The main question addressed by the research is to evaluate the binding and neutralizing antibody responses in individuals vaccinated with inactivated SARS-CoV-2 vaccines. The study aims to assess the consistency and correlation between binding antibodies (S-IgG and N-IgG) and neutralizing antibodies (Nabs) in individuals vaccinated with inactivated COVID-19 vaccines. For this purpose, the study addressing the gap in the field is original and relevant to the field of COVID-19 immunology.
Compared to previous studies, the presented study can be considered original in terms of the following aspects and also provides valuable information to the subject area.
· In addition to examining S-IgG, N-IgG and Nabs levels in vaccinated individuals, the correlation between these antibodies and their vaccine effectiveness was also investigated.
· The value of IgG titers in predicting the positivity of neutralizing antibodies, which is a critical issue in vaccine evaluation and epidemiological research, has been identified. This provides practical information for assessing the immune response to COVID-19.
Overall, this study offers a comprehensive analysis that enhances our comprehension of vaccine-induced immune responses and their significance for COVID-19 control and prevention. It also uncovers novel insights into the correlation between binding and neutralizing antibodies in the context of inactivated SARS-CoV-2 vaccine vaccination.
The conclusions drawn in the research study appear to be consistent with the evidence and arguments presented throughout the paper. The conclusions address the main question posed in the study, which is to assess the relationship between S-IgG, N-IgG, and Nabs in vaccinated individuals and to determine the predictive value of IgG titers in identifying the seroconversion of Nabs. Overall, the conclusions drawn in the research study align with the evidence presented and effectively address the main question posed regarding the evaluation of binding and neutralizing antibodies in individuals vaccinated with inactivated SARS-CoV-2 vaccines.
The references provided in the analysis are appropriate and relevant to the discussion of the research study on the evaluation of binding and neutralizing antibodies in individuals vaccinated with inactivated SARS-CoV-2 vaccines. The references cited include primary research articles that support the findings and conclusions presented in the analysis.
Round 2
Reviewer 1 Report
Comments and Suggestions for Authors
Accepted in this form. Good work
Author Response
We thank you for your helpful review and comments to our manuscript.